# Non-Gaussian Tensor Programs

**Eugene Golikov**
École Polytechnique Fédérale de Lausanne
Lausanne, Switzerland
evgenii.golikov@epfl.ch

**Greg Yang**
Microsoft Research
Seattle, USA
gregyang@microsoft.com

## Abstract

Does it matter whether one randomly initializes a neural network (NN) from Gaussian, uniform, or other distributions? We show the answer is "yes" in some parameter tensors (the so-called *matrix-like* parameters) but "no" in others when the NN is wide. This is a specific instance of a more general *universality principle* for Tensor Programs (TP) that informs precisely when the limit of a program depends on the distribution of its initial matrices and vectors. To obtain this principle, we develop the theory of non-Gaussian Tensor Programs. As corollaries, we obtain all previous consequences of the TP framework (such as NNGP/NTK correspondence, Free Indepedence Principle, Dynamical Dichotomy Theorem, and $\mu$-parametrization) for NNs with non-Gaussian weights. [1]

## 1 Introduction

**Universality in Neural Network Initialization**   Common initialization schemes of neural networks (e.g., [6, 11]) define specific ways of scaling initial layer weights with layer sizes. However, in general, they do not express any preference on the initial weight distribution beyond iid sampling. While some works [13, 12] use a Gaussian distribution, other practitioners advocate for a uniform one since it is possible to sample very large weights from a Gaussian, causing numerical instabilities.

However, a theorist, especially with a background on physics or probability theory, would suspect that the precise choice of distribution does not matter as the neural network's width tends to infinity. This is a belief in the general concept of *universality* — large systems display a consistent behavior at a macroscopic scale regardless of the microscopic details.[2]

This consideration has led deep learning practitioners to treat the distribution of random initialization as a matter of personal choice. However, as we shall see, the reality appears to be more subtle: roughly speaking, distribution universality holds for hidden weights but does not hold for other parameters like input and output weights. We formulate the precise *universality principle* (Principle 2) for neural network random intialization in Section 2. We do so in plain English, but deducing this principle requires some fundamental advancements in the theory of *Tensor Programs.* Below, we briefly review this theory before describing our contributions to it.

**Tensor Programs**   Just like autograd [25] empirically automates the calculation of chain rule of arbitrary computation graphs, Tensor Programs (TP) [28] has automated the theoretical calculation of infinite-width limits of the same (where width of a computation graph corresponds to the size of matrices). What previously were difficult limits to calculate, now becomes routine via TP. For example, [17] in 1994 showed that randomly initialized wide shallow neural networks are Gaussian Processes (which is called the Neural Network-Gaussian Process Correspondence, or

---

[1]We refer the reader to the arXiv version for the latest version of the manuscript.

[2]The simplest example of universality: taking the average of many iid copies of a random variable always yield its mean, regardless of whether the variable is Gaussian, uniform, Laplace, etc.

36th Conference on Neural Information Processing Systems (NeurIPS 2022).

NNGP Correspondence), but only recently this has been extended to deep perceptrons [15, 16] and more advanced architectures such as convolutional neural networks [5, 18], and each such extension requires painstaking calculations and careful applications of Law of Large Numbers and Central Limit Theorem. But with Tensor Programs, one can show that NNGP Correspondence holds for any architecture all at once [29]. Similarly, in a certain parametrization, a wide multi-layer perceptron (MLP) evolves like a linear model during training [14], but showing this for advanced architectures was very difficult. TP [30, 34] again was able to prove this behavior for any architecture. Finally, TP gave rise to the Dynamical Dichotomy Theorem [32], a classification of all natural infinite-width limits of neural networks, and led to the discovery of Maximal Update Parametrization, or $\mu$P. These results underlie the hyperparameter transfer technology that for the first time enabled the hyperparameter tuning of enormous neural networks too expensive to train more than once [33].

**Distributional Universality**   However, these results were only proven for computational graphs whose matrices and vectors are random Gaussians (which, in the special case of graphs pertaining to neural network training, means random Gaussian initialization of NN). What happens when they take more general distributions?

In this work, we show that any program with non-Gaussian matrices and vectors also have infinite-width limits under mild conditions, and thus recovering all of the aforementioned results for non-Gaussian objects automatically (see Section 4). In fact, they often coincide with the limits of Gaussian samplings: we shall formulate a general *distributional universality principle* for Tensor Programs (Principle 3), from which the universality principle (Principle 2) for NN random intialization (mentioned in the beginning of this section) follows as a special case.

For those familiar with the language of Tensor Programs, the principle can be summarized simply: all programs have the same limit if their Gaussian matrices are swapped out for non-Gaussian matrices with the same variances; however, this is not true in general for their initial vectors.

**Applications to Random Matrix Theory**   The universality principle also makes the proof of the Semicircle and Marchenko-Pastur Laws in [31] automatically valid for non-Gaussian random matrix ensembles. Likewise, it shows that the asymptotic singular value distribution of the input-output Jacobian of a random neural network does not depend on the distribution of random initialization beyond its variance.

**Contributions**   In summary:

- We clarify where initialization distribution matters and where it does not in wide neural networks by formulating a precise *universality principle* (Principle 2).
- More generally, we develop the theory of non-Gaussian Tensor Programs, as well as stating the corresponding *universality principle* for Tensor Programs (Principle 3).
- We apply this general theory to obtain previous results of the Tensor Programs papers, such as NNGP and NTK correspondence, for non-Gaussian weight initializations (Section 4).

## 2   Distributional Universality in Words

In this section, we state, at a level understandable to practitioners, where the sampling distribution in random initialization matters (beyond the first two moments). We begin with the simple example of MLPs.

**Motivating Example: MLP**   Let us first consider the case of a simple biasless multilayer perceptron (MLP) $f(\xi)$ with $L$ hidden layers and a nonlinearity $\phi$:

$$f(\xi) = W^{L+1}\phi(W^L\phi(\cdots\phi(W^1\xi)\cdots)). \tag{1}$$

Here $W^2, \ldots, W^L \in \mathbb{R}^{n \times n}$, $W^1 \in \mathbb{R}^{n \times d_{\text{in}}}$ and $W^{L+1} \in \mathbb{R}^{d_{\text{out}} \times n}$, where $d_{\text{in}}$ (resp. $d_{\text{out}}$) is the input (resp. output) dimension, and we call $n$ the *width* of $f$. Then we can formulate the following *universality* principle:

**Principle 1** (Universality in MLP Initialization)**.** *As width tends to infinity, two different iid random initializations of a biasless multilayer perceptron (MLP) induce identical training behavior as long as*

*1) they sample input and output weights the same way, and 2) they sample hidden weights with the same mean and variance.*[3]

Here "identical training behavior" means that for *any* sequence of batches of data, performing SGD from either random initialization yields the same function after any number of training steps.[4]

The key point in Principle 1 is that hidden weights are not sensitive to the exact distribution but the input and output weights are. We can demonstrate the sensitivity of the input and output weights on a simple example. Consider the 1-hidden-layer (i.e. $L = 1$) version of Eq. (1) with $d_{\text{in}} = d_{\text{out}} = 1$ and with input and output weights tied, which we name $U = W^1 = W^{2\top} \in \mathbb{R}^n$ (where again $n$ is width):

$$f(\xi) = \frac{1}{n} U^\top \phi(U\xi). \tag{2}$$

Here, the additional $\frac{1}{n}$ factor compared to Eq. (1)[5] is just a normalization so that the network output will not blow up to infinity as width $n$ becomes large; it's convenient but not essential for what we will discuss. Consider two alternatives (G) and (R) for sampling $U$:

(R) $U_\alpha = \pm 1$ with prob. $1/2$  or  (G) $U_\alpha \sim \mathcal{N}(0,1)$.

Both methods have variance 1. Now suppose the nonlinearity $\phi(x)$ equals to $x\mathbb{I}[x \in -1/2, 1/2]$ Then

$$f(1) = 0 \text{ with init (R)} \quad \text{but} \quad f(1) \to \mathbb{E}_z z\phi(z) > 0 \text{ with init (G)}$$

as $n \to \infty$, where $z \sim \mathcal{N}(0,1)$. Thus (G) and (R) definitely do not induce identical training behaviors — they are not even identical at initialization!

This example can be generalized to deep biasless MLPs to show that input and output weights are sensitive to the sampling distribution. Conversely, from our main theorem (Theorem 3.7) below, it will also be clear that hidden weights are insensitive to the sampling distribution.

**What Principle 1 Gets Wrong in General Architectures**  Unfortunately, Principle 1 is not correct if we go beyond biasless MLP. Indeed, if we just add bias to such an MLP, then one can easily generalize the example of Principle 1 to show that biases are also sensitive to their exact sampling distribution. As the architecture becomes complex, it becomes difficult to figure out whether a particular parameter tensor is sensitive or not in an ad hoc fashion (e.g., layernorm weights and biases? Self-attention weights? etc).

**Principle for General Case**  Fortunately, there is a simple rule to tell which parameter tensors are sensitive based on the following.

**Definition 2.1** ([33]). Let $P$ be a parameter tensor in a neural network of any architecture. As width becomes large, if $P$'s size remains constant, then we say $P$ is *scalar-like*; if exactly one dimension of $P$ becomes large, we say $P$ is *vector-like*; if exactly two dimensions of $P$ become large, we say $P$ is *matrix-like*.[6]

**Example 2.2.** In Eq. (1), $W^2, \ldots, W^L$ are all matrix-like while $W^1$ and $W^{L+1}$ are vector-like because $d_{\text{in}}$ and $d_{\text{out}}$ are fixed even as $n$ varies.[7] If we add biases,

$$f(\xi) = b^{L+1} + W^{L+1}\phi(b^L + W^L\phi(\cdots \phi(b^1 + W^1\xi)\cdots)), \tag{3}$$

then $b^{L+1} \in \mathbb{R}^{d_{\text{out}}}$ is scalar-like while $b^1, \ldots, b^L \in \mathbb{R}^n$ are vector-like.

**Example 2.3.** Some more advanced examples for practitioners, that may be skipped on first reading: If $f$ in Eq. (3) is convolutional (instead of dense) then the same categorization (of $W^l, b^l$ into scalar-, vector-, and matrix-like) holds because the kernel size of convolutions is constant as width increases:

---

[3]"same" means between the two initialization methods; on the other hand, different weight entries can have different distributions.

[4]This can be made even more general in the context of Tensor Programs; see [34, 32].

[5]This results in the so-called mean field parametrization [24, 3, 26], which is a special case of the maximal update parametrization [32].

[6]We can further define *tensor-like* and so on, but practically speaking, all relevant large neural networks (e.g., BERT, GPT-3, etc [4, 1]) will have at most matrix-like parameters due to storage and computation costs of "tensor-like" parameters.

[7]Think of $d_{\text{in}}$ and $d_{\text{out}}$ as being fixed by the dataset, while you, as a model builder, can freely vary $n$.

1) $W^2, \ldots, W^L$ are matrix-like, 2) $W^1, W^{L+1}, b^1, \ldots, b^L$ are vector-like, and 3) $b^{L+1}$ is scalar-like. Layernorm weights and biases are vector-like if the input to that layernorm has exactly one hidden dimension (which is almost always the case in practice). Self-attention weights $W^k, W^q, W^v$ are matrix-like if $d_{head}$ is fixed as $d_{model}$ and $n_{head}$ increases or if $n_{head}$ is fixed as $d_{model}$ and $d_{head}$ increases (one of which happens almost always in practice).

With this concept of scalar-, vector-, and matrix-like tensors in mind, we can formulate the general universality principle for neural network random initialization.

**Principle 2** (Universality in General Neural Network Initialization). *As width becomes large, two different iid random initializations of a neural network of any architecture* induce identical training behavior[8] *as long as 1) they sample scalar- and vector-like parameters the same way, and 2) they sample matrix-like parameters with the same mean and variance.*

**Remark 2.4.** Note both Principle 1 and Principle 2 only give *sufficient* conditions for identical training behavior for *all possible datasets and batches*. But practically, when we only focus on specific datasets and specific training procedures at hand, it is possible that there could be weaker conditions that would ensure identical training behavior in such specific settings. For example, if the input dimension of a dataset is large and each example has entries that look iid then the input layer is not sensitive to the exact sampling distribution (aside from the first two moments) either. Nevertheless, Principle 1 and Principle 2 yield guidelines that are generally applicable, upon which we may refine our reasoning, if we wish, based on individual specifics.

## 3 Tensor Programs: Main Result

Colloquially, a Tensor Program is just a computation interleaving matrix multiplication and coordinatewise nonlinearities. In prior works, there have been several different formalizations of this concept. Here, we simplify and take the following definition.

**Definition 3.1.** Given matrices $A^1, \ldots, A^L \in \mathbb{R}^{n \times n}$, initial vectors $g^1, \ldots, g^{M_0} \in \mathbb{R}^n$, and initial scalars $c^1, \ldots, c^{M_0} \in \mathbb{R}$, consider the following iteration for $i = M_0 + 1, \ldots, M$ that generates new vectors $g^{M_0+1}, \ldots, g^M \in \mathbb{R}^n$ and scalars $c^{M_0+1}, \ldots, c^M \in \mathbb{R}$:

$$g_\alpha^i \leftarrow \sum_{\beta=1}^n W_{\alpha\beta}^i x_\beta^i, \quad c^i \leftarrow \frac{1}{n} \sum_{\beta=1}^n x_\beta^i, \quad \text{where } x_\alpha^i = \phi^i(g_\alpha^1, \ldots, g_\alpha^{i-1}; c^1, \ldots, c^{i-1}). \tag{4}$$

Here each $\phi^i$ is a chosen scalar function with $(i-1) + (i-1)$ arguments and $W^i$ is an $n \times n$ matrix. Each matrix $W^i$ equals to either some matrix $A^j$ of the program or its transpose $A^{j\top}$. The matrices $W^i$ for different $i$ can possibly be the same. In this work, we shall call any computation of this form a *Tensor Program (TP)*, or just a *program* for short. Thus each program is entirely determined by the data $\{A^j\}_{j=1}^L \cup \{g^i\}_{i=1}^{M_0} \cup \{c^i\}_{i=1}^{M_0} \cup \{\phi^i\}_{i=M_0+1}^M$ along with the correspondence between $W^i$ and $A^j$ or $A^{j\top}$.

This formulation of a Tensor Program is equivalent to NETSOR$\top^+$ in [31], as shown in Appendix E. As such, Eq. (4) can express any computation expressible in a DL framework such as PyTorch [20], including gradient descent iterations of neural networks of any architecture, e.g. [32, 34]. This expressivity allows one to treat a wide range of problems uniformly using just Theorem 3.4 below.

**Example program** For example, consider the first forward pass of a simple MLP with scalar input $\xi$ and output $f(\xi)$:

$$f(\xi) = V^\top \sigma(W \sigma(\xi U)), \tag{5}$$

where $\xi \in \mathbb{R}; U, V \in \mathbb{R}^n; W \in \mathbb{R}^{n \times n}$, and $\sigma$ is an activation function. We can express this in a TP as follows: $g^1 \leftarrow U, g^2 \leftarrow nV$ are the initial vectors and $c^1 \leftarrow \xi, c^2 \leftarrow 1$ are initial scalars (where $c^2$ will just be ignored). $A^1 \leftarrow W$ is the sole matrix of the program. Then the program computes

$$g^3 \leftarrow W^3 x^3, \quad \text{where } W^3 \leftarrow A^1 = W \text{ and } x_\alpha^3 = \phi^3(g_\alpha^1, g_\alpha^2; c^1, c^2) \leftarrow \sigma(\xi U_\alpha)$$

$$f(\xi) = c^4 \leftarrow \frac{1}{n} \sum_{\beta=1}^n x_\beta^4, \quad \text{where } x_\alpha^4 = \phi^4(g_\alpha^1, g_\alpha^2, g_\alpha^3; c^1, c^2, c^3) \leftarrow (nV)_\alpha \sigma(g_\alpha^3)$$

---

[8] See the discussion below Principle 1 for the meaning of *identical training behavior*.

where $c^3$ and $g^4$ are implicitly computed but ignored.[9] Extending this first step, the entire training process can further be written in a TP, where the learned function outputs are expressed as scalars. See [29, 30, 34, 32] for more examples.

**Gaussian Tensor Programs**    The results achieved by the TP framework [28, 31, 34] so far most commonly spring from the following version of the so-called *Master Theorem*:

**Theorem 3.2** (Gaussian Master Theorem, original formulation of [31]). *Consider Setup 3.3 below. Then, as $n \to \infty$, for any pseudo-Lipschitz $\psi$,*

$$\frac{1}{n} \sum_{\alpha=1}^{n} \psi\left(g_\alpha^1, \ldots, g_\alpha^M, c^1, \ldots, c^M\right) \xrightarrow{a.s.} \mathring{\Psi}, \tag{6}$$

*where $\mathring{\Psi}$ is a deterministic scalar given by a certain recurrent formula.*

**Setup 3.3** (Gaussian Tensor Programs). *Consider a Tensor Program with $M$ vectors $g^1, \ldots, g^M \in \mathbb{R}^n$ and scalars $c^1, \ldots, c^M$. Suppose 1) all initial vectors $g^1, \ldots, g^{M_0}$ have standard Gaussian entries[10];2) all initial scalars $c^1, \ldots, c^{M_0}$ have almost sure limits as $n \to \infty$; 3) all matrices $A^i$ have iid entries from $\mathcal{N}(0, n^{-1})$; 4) all the nonlinearities $\phi^i$ are pseudo-Lipschitz.[11]*

For example, Theorem 3.4 implies that Eq. (5)'s function values $f(\xi)$ *after training* converge to deterministic values in various infinite-width limits, in particular, the feature learning limit [32, 33].

We can reformulate the above theorem in a shorter form:

**Theorem 3.4** (Gaussian Master Theorem, equivalent formulation). *Consider Setup 3.3. Then, as $n \to \infty$, every scalar $c^i$ converges almost surely to a deterministic limit $\mathring{c}^i$ which can be computed via a recurrent formula.*

It is easy to see that the above statement is equivalent to the original Theorem 3.2. Indeed, any scalar in the program has the form $\frac{1}{n} \sum_\alpha \psi(g_\alpha^1, \ldots, g_\alpha^M; c_\alpha^1, \ldots, c_\alpha^M)$ for some function $\psi$, while any expression of the above form can be thought as a scalar in a new program. We are going to build upon the formulation of Theorem 3.4 since it introduces fewer entities.

**Non-Gaussian Tensor Programs**    We are going to generalize Theorem 3.4 to non-Gaussian distributions. But before we do so rigorously, we first formulate an easily statable principle that summarizes our results in an intuitive way.

**Principle 3** (Universality in Tensor Program Sampling). *For simplicity, consider TP without initial scalars. As $n \to \infty$, two different iid random samplings of a TP's matrices and initial vectors result in identical limits of scalars as long as 1) they sample all initial vectors the same way, and 2) they sample all matrix entries with the same variance (and zero mean).*

More generally, for programs with initial scalars, we just require the two samplings to have the same almost sure deterministic limits for them.[12]

Principle 1 and Principle 2 are special cases of Principle 3, since any neural network can be expressed as a TP. Principle 3 itself will follow straightforwardly from Theorem 3.7.

Now, let us setup our discussion of our main theorem, Theorem 3.7.

**Definition 3.5.** We say $f : \mathbb{R}^k \to \mathbb{R}$ is *polynomially smooth* if it is $C^\infty$ and its partial derivatives of any order are polynomially bounded, i.e. for any sequence $(P_1, \ldots, P_r) \in [k]^r$, we have $\left|\frac{\partial^r}{\partial x^{P_1} \ldots \partial x^{P_r}} f\right| \leq C(1 + |x_1|^p + \cdots + |x_k|^p)$ for some $C, p > 0$ that may depend on $(P_1, \ldots, P_r)$.

---

[9]We intentionally simplified the formulation of TP to the form Eq. (4) in particular to simplify the proofs, but at the cost that expressing common computation encounters some redundancy as shown in this example.

[10]In the original formulation of [31], initial vectors were assumed to be sampled as $(g_\alpha^1, \ldots, g_\alpha^{M_0}) \sim \mathcal{N}(\mu^{in}, \Sigma^{in})$ iid over $\alpha = 1, \ldots, n$ for some $\mu^{in} \in \mathbb{R}^{M_0}, \Sigma^{in} \in \mathbb{R}^{M_0 \times M_0}$. However, one can always construct such vectors from ones with iid standard Gaussian entries using a linear elementwise map. This map can be further absorbed into subsequent nonlinear maps in the program.

[11]A function $f : \mathbb{R}^k \to \mathbb{R}$ is called *pseudo-Lipschitz* if $|f(x) - f(y)| \leq C\|x - y\|(1 + \sum_{i=1}^{k} |x_i|^d + |y_i|^d)$ for some $C, d > 0$.

[12]We can, in fact, allow the initial scalars to have non-deterministic limits, in which case the non-initial scalars will have non-deterministic limits as well.In this case, we want the two samplings to have identical limit distributions for the joint distributions of all initial scalars.

**Setup 3.6.** *Consider Setup 3.3, but replace 3) and 4) with the following: 3\*) there exists a sequence $\nu_3, \nu_4, \ldots > 0$ such that all matrices $A^i$ have independent entries drawn from distributions with zero mean, variance $n^{-1}$, and all higher $k$th moment bounded by $\nu_k n^{-k/2}$; 4\*) all the nonlinearities $\phi^i$ are polynomially smooth. We further require 5) all moments of initial scalars $c^1, \ldots, c^{M_0}$ to exist.*

Many widely-used nonlinearities, such as ReLU or MaxPool, are not polynomially smooth, thus contradicting 4\*). Nevertheless, if the original nonlinearity is continuous and polynomially bounded (in particular, if it is pseudo-Lipschitz), one may apply Gaussian smoothing and get a polynomially smooth nonlinearity; see Appendix B. The narrower the smoothing kernel, the better the approximation.

The most relevant specific scenario satisfying 3\*) is if $A^i_{\alpha\beta} \sim \frac{1}{\sqrt{n}}\mathcal{D}$, where $\mathcal{D}$ is a fixed distribution, like uniform or truncated Gaussian with mean 0 and variance 1. In this case, $\nu_k$ can just be taken to be the $k$th moment of $\mathcal{D}$. But in general 3\*) allows entries of $A^i$ to come from different distributions.

Our main result is the following:

**Theorem 3.7** (Non-Gaussian Master Theorem). *Consider Setup 3.6. Then, as $n \to \infty$, every scalar $c^i$ converges to the same $\mathring{c}^i$ as in Theorem 3.4 almost surely and in $L^p$ for every $p \in [1, \infty)$:*

$$c^i \xrightarrow{\text{a.s. \& } L^p} \mathring{c}^i, \quad \forall p \in [1, \infty).$$

In short, Theorem 3.7 relaxes matrix sampling to be non-Gaussian and non-identically-distributed in general, at the cost of requiring a) more smoothness in nonlinearities and b) all moments of initial scalars to exist. In past applications of TP, b) has always been satisfied but a) has not for relu networks. On the other hand, we also gain $L^p$ convergence compared to Theorem 3.4.[13] Note that Setup 3.6 still requires initial vectors to be Gaussian, but this is not essential, as we discuss in Section 4.

**On Tensor Programs with variable dimensions.** While Setup 3.3 and 3.6 assume all hidden dimensions to be equal, Theorem 3.4 holds also for Tensor Programs with variable dimensions, see e.g. [31]. Since our proof technique is based on interpolation between Gaussian and non-Gaussian weights, it can be straightforwardly extended to variable dimensions, as long as Theorem 3.4 holds in this setting.

## 4 Applications

As mentioned in the introduction, the Tensor Programs series of papers so far has proven a wide range of results, which typically have the characteristic of *architectural universality*, i.e. covering most existing neural architectures. But all of such results have assumed Gaussian weight initialization. Now, armed with Theorem 3.7, we show that the same results hold with *non-Gaussian* weight initialization as well under mild assumptions, thus extending them to other prevalent initializations such as uniform and truncated Gaussian. Theorem 3.7 here acts like a drop-in replacement for Theorem 3.4 in their proofs, except we need to add more smoothness assumptions on nonlinearities.

In general, coarse, qualitative statements, such as "wide neural networks at initialization are Gaussian processes," still hold as stated. But note the exact quantitative statement, in this example regarding the kernel of the Gaussian process, can change as one changes the sampling distribution of vector-like and scalar-like parameters, as indicated by Principle 2 and Principle 3

**Definition 4.1.** We shall call a scalar random variable a *Gaussian image* if it is an image of a standard Gaussian variable under a polynomially smooth function.

Below, we express a rather abstract notion of a neural network that may seem bewildering to a reader who has not read previous papers in the Tensor Programs series. It helps to keep in mind specific examples, for example, the MLP with biases from Section 2, when reading the below statements. This notion generalizes all typical neural architectures (as shown in [29]) whose matrix-like parameters are randomly initialized with mean 0 and variance $\Theta(1/n)$ (under mild moment conditions of Setup 3.6) and whose scalar- and vector-like parameters are randomly initialized from a Gaussian image.

---

[13]We believe $L^p$ convergence can be established without requiring so much smoothness in nonlinearities, but leave this to future work.

**Setup 4.2.** *Consider a neural network $f(\xi) = \frac{1}{\sqrt{n}} V \Phi(\xi)$ with output layer weights $V$ and embedding $\Phi(\xi)$ on input $\xi$, and where $n$ is the dimension of the Tensor Program we describe next. Suppose 1) $\Phi(\xi)$ can be expressed as a concatenation of a constant (wrt $n$) number of vectors $x^i \in \mathbb{R}^n$ (c.f. Eq. (4)) from a Tensor Program $T$ in Setup 3.6, and 2) $V$'s entries are initialized iid from a Gaussian image with mean 0 and variance 1 and are independent from the random objects of the program $T$.*

**NNGP Correspondence.** The result below follows from Theorem 3.7:

**Corollary 4.3.** *Any neural network function described in Setup 4.2* [14] *converges in finite-dimensional distribution to a Gaussian process in the limit of $n \to \infty$.*[15]

In Appendix D, we prove a similar result without assuming the activation functions to be polynomially smooth (we require Lipschitzness instead), see Corollary D.3. This result will follow from our non-Gaussian Lipschitz Master Theorem, Theorem D.2. The proof of the above corollary follows the same lines as the proof of Corollary D.3 given in Appendix D[16].

As discussed above, the kernel of this process in general will be affected by the distribution of vector-like parameters. For example, take $f(\xi) = 1/\sqrt{n} V \phi(U\xi)$ in Setup 4.2 where $\xi \in \mathbb{R}$ and $\phi$ is the indicator function of the interval $[-1/2, 1/2]$. Then sampling $U_\alpha$ as $\pm 1$ with equal probability implies that $f$ is identically 0 while sampling $U_\alpha$ from a standard Gaussian implies $f$ converges to a nontrivial Gaussian process.

**NTK Correspondence.** We can directly plug our Theorem 3.7 into the proof of [34] and obtain

**Corollary 4.4.** *Consider Setup 4.2 and assume the loss function is continuously differentiable in the network output. Then under NTK parametrization and SGD weight updates*[17]*, in the limit of $n \to \infty$, 1) the NTK of the network at any optimization step converges pointwise almost surely to a finite deterministic limit $\mathring{\Theta}$ that does not depend on the timestep, and 2) moreover, the network function evolves according to kernel gradient descent with kernel $\mathring{\Theta}$.*

As in the NNGP case, the NTK's infinite-width limit can also be affected by the distribution of input and output weights. The same example there illustrates this: the network's limit NTK as $\phi$ is equal to the kernel of the limit Gaussian process because $\phi$ has 0 derivative almost everywhere.

We empirically validate Corollaries 4.3 and 4.4 in Appendix N.

While our non-Gaussian Lipschitz Master Theorem, Theorem D.2, mentioned above allows us to generalize Corollary 4.3 to ReLU nets, it does not allow us to generalize Corollary 4.4 in the same setup (because a Tensor Program expressing the backward pass of a ReLU net has ReLU derivatives, which are not even continuous, as nonlinearities).

**Random Matrix Theory.** Our Theorem 3.7 implies the semi-circle law for non-Gaussian Wigner matrices, the Marchenko-Pastur law for $AA^\top$, where $A$ is non-Gaussian, and Free Independence Principle (FIP) for Tensor Programs with non-Gaussian initial weights, thus generalizing TP3 [31]; we discuss these results in Appendix C. Moreover, since our Theorem 3.7 guarantees convergence in mean, we were able to state FIP without assuming linearly bounded nonlinearities as in [31].

---

[14]As we can notice from analyzing the proof of Corollary D.3, we do not need the output weights $V$ to be Gaussian images. Instead, we could assume that they are iid with zero mean, unit variance, and all higher moments existing. However, we are not able to prove Corollary 4.4 below without assuming the entries $V$ to be Gaussian images. We assumed a weaker setup since we wanted to have the same setup for both corollaries.

[15]Notice we allow a weight matrix and its transpose be both involved in the forward pass of the network, in contrast to [29], but in line with a more general theorem in [31].

[16]There is, however, a small subtlety when applying the proof technique of Corollary D.3 to Corollary 4.3. There we use a test function $\psi$ and take it to be Lipschitz and bounded. Theorem D.2 used to prove Corollary D.3 works for Lipschitz nonlinearities, therefore we could embed $\psi$ into the underlying tensor program. However, Theorem 3.7 does not work for Lipschitz nonlinearities and it is not obvious if we can use polynomially smooth functions as test functions for weak convergence. Nevertheless, we can take a bounded Lipschitz test function $\psi$ and smoothen it with a kernel of width $\delta > 0$ the same way as we do in the proof of Theorem D.2. Then it is easy to see that $|c - c^\delta| \le \delta$ surely for any $\delta > 0$, where $c^\delta$ is the same as $c$ in the proof of Corollary D.3 but with a $\delta$-smoothed version of $\psi$. Since $|\mathbb{E} c - \mathring{c}| \le |\mathbb{E} c - \mathbb{E} c^\delta| + |\mathbb{E} c^\delta - \mathring{c}^\delta| + |\mathring{c}^\delta - \mathring{c}|$, we get $\limsup_{n \to \infty} |\mathbb{E} c - \mathring{c}| \le \delta + |\mathring{c}^\delta - \mathring{c}|$. Taking infimum over $\delta > 0$ gives $\mathbb{E} c \to \mathring{c}$.

[17]For the exact meaning of this setup, see [34].

**Classification of Infinite-Width Limits.** Consider now an $L$-hidden-layer biasless perceptron with width $n$ trained using stochastic gradient descent (SGD). As in [32, Sec 3.2], we shall assume in this section that this perceptron's nonlinearities are either tanh or the so-called $\sigma$-gelu for sufficiently small $\sigma$ (c.f. [32, Assm 3.1]). [18] Note both tanh and $\sigma$-gelu are polynomially smooth.

We generalize the notion of *abc-parametrization* from [32] to non-Gaussian initializations:

**Definition 4.5.** Let $\mathcal{D}$ be a distribution with mean 0, variance 1, and all moments finite. An *abc-parametrization for $\mathcal{D}$*, specified by a set of numbers $\{a_l, b_l\}_{l=1}^{L+1} \cup \{c\}$, parametrizes the MLP as follows: 1) each weight factors as $W^l = n^{-a_l} w^l$ for the actual trainable parameter $w^l$, 2) the weights are sampled iid $w_{\alpha\beta}^l \sim n^{-b_l}\mathcal{D}$ at initialization, and 3) the SGD learning rate is taken as $\eta n^{-c}$ for some constant $\eta$.

In their Dynamical Dichotomy Theorem, [32] classified all abc-parametrizations where $\mathcal{D} = \mathcal{N}(0,1)$ into the following categories: stable, nontrivial, kernel regime, and feature learning. Here, the *stable* and *nontrivial* categories overlap, and the *kernel* and *feature learning* regimes are mutually exclusive categories within their intersection. Each category is characterized by some set of linear inequalities in $\{a_l, b_l\}_{l=1}^{L+1} \cup \{c\}$ (c.f. [32, Sec 3]). It turns out that all but one abc-parametrization, the so-called $\mu$-parametrization (abbreviated $\mu P$), exhibit defects in the infinite-width limit (such as losing the ability to learn features). This is formalized in [32, Thm 5.6]. [33] then showed that $\mu P$ gives rise to a new technology called $\mu$*Transfer* that allows one to, for the first time, tune extremely large neural networks too expensive to train more than once, such as GPT-3 [1].

Using Theorem 3.7, we see that all of the above notions, originally defined for Gaussian initialization, in fact are distributionally universal:

**Theorem 4.6.** *Let $\mathcal{D}, \mathcal{D}'$ be two distributions satisfying the moment conditions of Definition 4.5. Then an abc-parametrization for $\mathcal{D}$ is in feature learning (resp. kernel/stable/nontrivial) regime iff it is so for $\mathcal{D}'$. It is the $\mu$-parametrization for $\mathcal{D}$ iff it is so for $\mathcal{D}'$.*

This suggests that the $\mu$Transfer technique mentioned above works regardless of the initialization distribution of weights.

**On potential societal impacts.** Our work concerns generic behavior of neural nets in the limit of infinite width and therefore does not provide any foreseen societal impacts. The only direct practical application of our work we are aware of is theoretical justification for the hyperparameter tuning method of TP5 [33] for non-Gaussian weight initializations. However, what our work provides is merely justification for this method, while the method itself existed before our work (and was well-justified for Gaussian weight initializations).

## 5   Proof of Theorem 3.7

**Limitations of the Proof Technique of Theorem 3.4.** The proof of Theorem 3.4 as given in [31] uses the Gaussianity of the matrices $A^j$ in a very essential way. It leverages the property of multivariate Gaussians to remain Gaussian after conditioning on *linear* constraints.

In particular, it lets one understand the distribution of $g^i$ conditioned on $g^{i-1}, \ldots, g^1$. Indeed, under such conditioning, $W^i$ is in general no longer iid but nevertheless constrained only by the following equalities (where now $g^j$ and $x^j$ should be considered deterministic under such conditioning):

$$g^j = W^j x^j, \quad \text{for all } j < i \text{ and where } W^j = W^i \text{ or } W^{i\top}.$$

This constraint is *linear* in $W^i$ (even though it's generally nonlinear in $g^1, \ldots, g^{i-1}$ since $x^j$ is a nonlinear function of them). Therefore, when conditioned on $g^{i-1}, \ldots, g^1$, the matrix $W^i$ is *still Gaussian*, albeit with some conditional mean and covariance. Quickly glossing over the rest of the proof, this allows one to reason about the conditional distribution of $g^i$ and eventually to reduce the almost sure convergence of the $(i+1)$th scalar $c^{i+1}$ to the almost sure convergence of the $i$th scalar in a different program. Then an inductive argument over $M$ for all programs of size $M$ can prove Theorem 3.4. A more detailed proof sketch can be found in [31, Section 6].

Now, when the matrices $W^i$ are no longer Gaussian, this argument completely breaks down as we no longer have good control of the conditional distribution of $W^i$ for general entry distributions. We

---

[18]This means $\sigma$-gelu smoothly approximates relu sufficiently well.

therefore apply a different argument. The idea is to interpolate the weights from the Gaussian ones, for which convergence of $c^i$ is given by Theorem 3.4, to the non-Gaussian ones, and show that $c^i$ does not change along this interpolation in the limit of large $n$.

**Our Proof of Theorem 3.7**

**Definition 5.1** (Interpolated Program). Given a program $T$ in Setup 3.6, let $\tilde{A}^i$ denote an iid Gaussian matrix with the same mean and variance as the non-Gaussian matrix $A^i$ of the program. Then we define the *interpolated program* $T(t)$ for $t \in [0, 1]$ as follows: $T(t)$ is identical to $T$ except that its matrices take the following values:

$$A^i(t) \stackrel{\text{def}}{=} \tilde{A}^i \cos\left(\frac{\pi}{2}t\right) + A^i \sin\left(\frac{\pi}{2}t\right), \quad \text{for all } t \in [0, 1] \text{ and } i = M_0 + 1, \ldots, M, \qquad (7)$$

Naturally, $W^i(t)$ inherits the same definition. The vectors and scalars in the program will change continuously as $t$ varies, and consequently we write them as $g^i(t)$ and $c^i(t)$ for $i = 1, \ldots, M$.[19]

In Section 5.1, we will discuss why this specific form of interpolation[20] is important in our case.

Here $t = 0$ corresponds to the Gaussian program, while $t = 1$ corresponds to the "target" non-Gaussian one. Eventually, we aim to show that all scalars $c^i(t)$, almost surely in the limit of $n \to \infty$, will remain constant as $t$ varies from 0 to 1 (as stated by Theorem 5.2 shortly after setting some notations), thus proving that the Gaussian and non-Gaussian programs have the same limits.

**Notations**    For any object (matrix, vector, or scalar) $\omega(t)$ of this interpolated program, we shorthand

$$\dot{\omega}(t) \stackrel{\text{def}}{=} \frac{d}{dt}\omega(t).$$

Big-O notation, e.g., $O(n^{-1/2})$, always suppress multiplicative constants independent of $n$, but which may depend on everything else. The supremum $\sup_t$ is always taken over $t \in [0, 1]$.

**Theorem 5.2.** *Consider a program in Setup 3.6 and let $c$ be a scalar in it. For any $p \geq 1$, we have*[21]

$$\sup_t \mathbb{E} \, |\dot{c}(t)|^p = O(n^{-p/2}), \quad \text{as } n \to \infty.$$

In other words, this result says $\dot{c}(t)$ is small in $L^p$ norm uniformly over all $t$. With Theorem 5.2, some routine calculations (detailed below) then show that $c(1)$ converges almost surely and in $L^p$ to the same limit as $c(0)$ yielding a proof of Theorem 3.7 as desired. Theorem 5.2 is proven in Appendix L, but we shall demonstrate it on a simple example in Appendix A.

**Routine Calculations Finishing the Proof of Theorem 3.7 from Theorem 5.2**    Once Theorem 5.2 is proven, we have, for any $p \geq 1$,

$$\mathbb{E} \, |c(1) - c(0)|^p = \mathbb{E} \, \left| \int_0^1 \dot{c}(t)dt \right|^p \leq \int_0^1 \mathbb{E} \, |\dot{c}(t)|^p \, dt = O(n^{-p/2})$$

where the inequality follows from power mean or Hölder's inequality and the last equality follows from Theorem 5.2. Since the RHS goes to 0 with $n$, this implies that $c(1) - c(0)$ converges to 0 in $L^p$ as $n \to \infty$. With a standard application of Borel-Cantelli lemma, this also implies almost sure convergence of $c(1) - c(0)$. Since $c(0)$ converges to $\mathring{c}$ almost surely by the Gaussian theorem, Theorem 3.4, the same holds for $c(1)$.

In the process of proving Theorem 5.2, we will have proved $p$th moment bound on $c(1)$ for every finite $p \geq 1$ (Lemma J.6). Then a standard truncation technique uses this to convert the almost sure convergence of $c(1)$ to $L^p$ convergence for all $p \in [1, \infty)$; see Theorem M.3.[22]

It remains to prove Theorem 5.2, which we demonstrate on a simple example in Appendix A after discussing the key properties of our interpolation (Eq. (7)).

---

[19]but note that they are invariant to $t$ for any $n$ when $i \leq M_0$ because by construction, the initial vectors and scalars are the same between the Gaussian and non-Gaussian programs.

[20]This interpolation is similar to that used in [2] in the context of approximate message passing, which can be considered as an application of the TP framework to a specific kind of programs; see Appendix F for comparison.

[21]The hidden multiplicative constant here is only independent of $n$ but can depend on $p$ and other details of the program.

[22]Note since $c(0)$ (the scalar in the Gaussian program) is not guaranteed by Theorem 3.4 to converge in $L^p$, the above $L^p$ convergence of $c(1) - c(0)$ does not immediately imply the $L^p$ convergence of $c(1)$.

## 5.1 Key Properties of Our Matrix Interpolation

There are may ways to interpolate between (or, more generally, to *couple*) $\tilde{A}$ and $A$ other than Eq. (7). Why did we pick Eq. (7)? The following lemma proven in Appendix G summarizes the key properties of it, where especial attention should be paid to the fact that $\mathbb{E}\, a\dot{a} = 0$:

**Lemma 5.3** (Interpolation Properties). *For any matrix entry* $a(t) = A^i_{\alpha\beta}(t)$ *of a program in Setup 3.6: (1)* $\mathbb{E}\,\dot{a}(t) = \mathbb{E}\,a(t)\dot{a}(t) = 0$ *for all* $t$; *(2) For any integers* $j, k \geq 0$ *with sum* $\ell = j + k$, $\sup_t \mathbb{E}\,|a(t)^j \dot{a}(t)^k| \leq \pi^\ell \nu_\ell n^{-\ell/2}$, *where* $\nu_\ell$ *is the scaled moment bound in Setup 3.6.*

The fact $\mathbb{E}\, a\dot{a} = 0$ is crucial and the main reason we picked the specific form Eq. (7) of the interpolation. It will allow us to kill leading terms in the Taylor expansions important to proving Theorem 5.2. We demonstrate this in an example in Eq. (11). On the other hand, (2) is just a very strong version of the obvious statement that "$a$ and $\dot{a}$ both have typical size $1/\sqrt{n}$," but importantly, this is uniform over $t \in [0, 1]$.

## 6 Related works

The Tensor Programs series discusses different applications of the (Gaussian) Master theorem proven by [28]. They include: Gaussian process behavior at initialization (TP1, [29]), convergence to a kernel method (TP2, [30]), Free Independence Principle (TP3, [31]), dynamical dichotomy and $\mu$-parameterization (TP4, [32]), and finally application of $\mu$P to hyperparameter tuning (TP5, [33]).

Neural networks converge to Gaussian processes as their width goes to infinity, as was proven by [15, 16] for fully-connected nets, and by [18, 5] for convolutional nets; see also [9]. Using the Master theorem, [29] showed that this behavior holds for a very wide class of architectures, including not only convolutional, but also graph convolutional and recurrent neural nets, ResNets, networks with batch normalization, and networks with attention. The seminal work of [14] demonstrated that under certain parameterization, the learning dynamics a neural net converges to that of a kernel method. The corresponding kernel was called Neural Tangent Kernel, or NTK, and drawn a lot of attention in recent years. While the result of [14] was proven only for fully-connected nets with smooth activations, the Master theorem allows to generalize this result for a wider class of architectures (the same as mentioned above), see [30]. [23, 21, 27] and others studied trainability of very deep and wide neural networks using random matrix theory. Their analysis crucially relied on the assumption that hidden representations of a neural network at initialization were *freely independent* from the weights. TP3 [31] was among the first works to validate this assumption rigorously; see also [19]. Infinite-width behavior of a neural net depends on scaling of its hyperparameters (like initial weights variance and learning rate) with width. Dynamical dichotomy proposed in TP4 [32] is a classification of scalings that are meaningful in a certain sense. Another classification of scalings with a different notion of meaningfulness was proposed earlier by [7, 8], but only for two-layered networks.

A distribution universality property similar to our Theorem 3.7 was shown by [2] for approximate messaging passing. However, their model does not cover most of possible neural network computations; see Appendix F for discussion.

## 7 Limitations of our results

First, our Theorem 3.7 is applicable only to Tensor Programs with smooth nonlinearities, which rules out several popular activation functions like ReLU or MaxPool. Our Theorem D.2 (see Appendix D) does not really solve the issue since a Tensor Program expressing the backward pass involves derivatives of the activation functions, which are not even continuous for ReLU. As a workaround, we could consider their smoothed versions (e.g. Softplus instead of ReLU) with a controllable smoothness parameter, and put this parameter very close to zero, thus getting "almost ReLU".

## 8 Conclusions

We present a generalization of the Master theorem of [28] to non-Gaussian weight initializations. Our generalization allows for the same applications as the original Master theorem, thus broadening the scope of applicability of the Tensor Programs machinery.

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
