# OpenReview forum: "Non-Gaussian Tensor Programs"
_NeurIPS.cc/2022/Conference — NeurIPS 2022 Accept_

### Official Review · Reviewer_eCbt · 2022-07-11

**Rating:** 6
**Confidence:** 3
**Soundness:** 4 excellent
**Presentation:** 3 good
**Contribution:** 3 good

**Summary:**

This work continues in the spirit of Tensor Program series of papers. Starting from a theoretical
result showing that nearly any neural architecture with Gaussian weight initialisation in the limit
of infinite width exhibits Gaussian Process behaviour, this paper extends this theorem to apply where
weights have non-Gaussian initialisation. This requires assuming that all matrices (A) used have iid
entries from a Gaussian distribution, all bias vectors also come from an iid Gaussian distribution, and
that all non-linearities be polynomially-smooth.


**Questions:**

  The paper offers to me seems a fairly modest contribution. With the restrictions needed to not require Gaussian
  weights it's unclear why simpler extensions could not have been done. For example, Appendix section A. states that
  images of smooth polynomial maps can be used to get around the Gaussian initial bias and layer weights.
  Why could the same trick not been used in original formulation?


**Limitations:**

I believe the authors have adequately addressed the limitations and potential negative societal impacts of the work.

**Strengths And Weaknesses:**

  Originality:

  This work is very original and I am not aware of any other theoretical contributions of this sort
  in the literature.

  Technical Quality:

  The work is technically sound and although I did not carefully inspect the proofs I don't have
  significant reasons to doubt the results. Although there are not experiments in the paper, given
  the theoretical nature of the contribution I don't think it's necessary.

  Clarity:

  The work relies heavily on familiarity with previous Tensor Programs papers for background and motivation
  which makes the paper and its contribution less self-contained. But given some familiarity, the paper was
  straightforward to read and follow.

  Significance:

  The paper offers to me seems a fairly modest contribution. With the restrictions needed to not require Gaussian
  weights it's unclear why simpler extensions could not have been done. For example, Appendix section A. states that
  images of smooth polynomial maps can be used to get around the Gaussian initial bias and layer weights.
  Why could the same trick not been used in original formulation?

---

> ### Author Response · Authors · 2022-08-01
> **Response**
>
> We would like to thank the anonymous reviewer for valuable comments!
> To begin, we address the main question posed:
>
> “With the restrictions needed to not require Gaussian weights it's unclear why simpler extensions could not have been done. For example, Appendix section A. states that images of smooth polynomial maps can be used to get around the Gaussian initial bias and layer weights. Why could the same trick not been used in original formulation?”
>
> The trick we used to convert Gaussian vectors to non-Gaussian ones cannot be applied to matrices simply because tensor programs do not allow applying an elementwise nonlinearity to an n x n matrix (A-variable). Indeed, the TP iteration (eq.(1)) allows for applying a nonlinearity elementwise to a finite set of vectors and to a final set of scalars, but it does not support applying it to an A-matrix.
>
> We have to stress out that input layer and output layer weights of a neural network are not expressed with A-matrices; instead, they are expressed with a set of vectors. For example, if a network has k input neurons and width n, its input layer weights are expressed as a set of k vectors. For all layers except for the input and the output ones, we cannot do the same because for a hidden layer with n input and n output units we would need n vectors - a number that obviously depends on n, which goes to infinity. That’s why we are able to map Gaussian input and output weights to non-Gaussian ones using a nonlinear function but we cannot apply any mapping to any other weight matrix.
>
> Second, we would like to correct several probable misconceptions we have noticed in the review:
>
> “Starting from a theoretical result showing that nearly any neural architecture with Gaussian weight initialisation in the limit of infinite width exhibits Gaussian Process behaviour, this paper extends this theorem to apply where weights have non-Gaussian initialisation.”
>
> We start with a stronger result from the previous work, Gaussian Master theorem (Theorem 1), from which Gaussian process behavior for Gaussian-initialized neural nets follows as a corollary. Our main result is non-Gaussian Master theorem, Theorem 2, from which Gaussian process behavior for non-Gaussian-initialized neural nets follows as a corollary, Corollary 1.
>
> The Master theorem is a much more general result then just Gaussian process behavior in the limit of inifinite width. First, a tensor program can express not only a forward pass of a neural network but also a backward pass and any number of gradient descent steps, while we need only the 1st forward pass to prove NNGP correspondence. Second, generally, vectors generated by a program do not exhibit a Gaussian process behavior; this is the case e.g. for programs expressing gradient steps. Nevertheless, vectors generated by a program expressing the very first forward pass indeed tend to Gaussians; this is NNGP correspondence, which is our Corollary 1 and the main topic of TP I [1].
>
> “This requires assuming that all matrices (A) used have iid entries from a Gaussian distribution, all bias vectors also come from an iid Gaussian distribution, and that all non-linearities be polynomially-smooth.”
>
> Concerning weight distributions, this is not true: we require all matrices in a tensor program to have iid entries from a distribution with zero mean, variance inversely proportional to width, and all higher moments existing; it does not have to be Gaussian, see Setup 2.
>
> When applying our result to neural nets, we express input and output layer weights as sets of vectors, while weights of all other layers are expressed as matrices. We need Gaussian output weights for Corollaries 1 and 2 to hold but all other weights can be non-Gaussian, see Setup 3.
>
> [1] Yang, G. (2019). Wide feedforward or recurrent neural networks of any architecture are gaussian processes. Advances in Neural Information Processing Systems, 32.

---

> > ### Comment · Reviewer_eCbt · 2022-08-08
> > **Re: Response**
> >
> > I want to thank the authors for their response and clearing up some of my misunderstandings about their work. The theoretical contribution is much stronger than I initially understood and I'll update my scores/review appropriately.

---

### Official Review · Reviewer_88YS · 2022-07-12

**Rating:** 5
**Confidence:** 1
**Soundness:** 3 good
**Presentation:** 2 fair
**Contribution:** 3 good

**Summary:**

The paper extends the Tensor Program to the non-Gaussian case, which justifies a previous hyperparameter tuning method for non-Gaussian weight initialization.

**Questions:**

The authors may consider moving the proofs to the appendix and using the space to explain more about Tensor Program and its application.

For example, in Section 2, readers may want to know how to convert a practical convolutional networks (with weight sharing convolutions and batch normalization) to Tensor Program. Since the authors emphasize on universality, it is also informative to discuss whether the scheme is applicable in graph neural networks, where the central limit theorem may not hold given sparsity.

Moreover, in Section 3, it is useful to discuss how to use the theoretical result for hyperparameter tuning, instead of citing pror works. Lastly, since the result only hold for infinite-width case, it is helpful to include some empirical study to show the gap between theory and practice.

**Limitations:**

Not applicable.

**Strengths And Weaknesses:**

Since I don't have a background in Tensor Programs, I can only comment on writing.

The proposed theory helps to explain some previous practices in learning neural networks; however, it does not suggest new techniques nor has empirical verification. Moreover, the paper is generally dense and dry, therefore challenging for non-specialists to follow.

---

> ### Author Response · Authors · 2022-08-01
> **Response**
>
> We thank the anonymous reviewer for valuable comments. We begin with addressing the concerns posed by the reviewer:
>
> "the paper is generally dense and dry, therefore challenging for non-specialists to follow."
>
> In the latest revision, we have added more clarifying remarks, improved notation, and got rid of the most of the multiline formulas in the main.
>
> “The authors may consider moving the proofs to the appendix and using the space to explain more about Tensor Program and its application.”
>
> The proofs are in the appendix already. Since Theorem 2 is the main result of our paper, in order to give the reader a gist of the proof technique, we decided to put a proof sketch for the simplest non-trivial tensor program possible, see Section 4.1. We discuss the applications in Section 3 and Appendix B.
>
> “In Section 2, readers may want to know how to convert a practical convolutional networks (with weight sharing convolutions and batch normalization) to Tensor Program. Since the authors emphasize on universality, it is also informative to discuss whether the scheme is applicable in graph neural networks, where the central limit theorem may not hold given sparsity.”
>
> Our work builds on the TP series, where representations of numerous practical architectures (including convolutional and graph neural nets) as tensor programs were discussed in detail, see e.g.  [1], Appendix A.
>
> “in Section 3, it is useful to discuss how to use the theoretical result for hyperparameter tuning, instead of citing pror works.”
>
> Our results do not suggest any new techniques. Instead, one of the corollaries of our main result is that the existing hyperparameter tuning technique is universal wrt initial weight distribution. Therefore the existing technique remains the same. Discussing the hyperparmeter tuning method of [2] would mean merely re-stating their method without any modifications.
>
> "Lastly, since the result only hold for infinite-width case, it is helpful to include some empirical study to show the gap between theory and practice."
>
> We have included some empirical results on NNGP correspondence and convergence of the initial NTK in Appendix I of the latest revision.
>
> We then focus on certain small misconceptions we have noticed in the review.
>
> "The paper extends the Tensor Program to the non-Gaussian case, which justifies a previous hyperparameter tuning method for non-Gaussian weight initialization."
>
> While justification of the mentioned hyperparameter tuning method is probably the main practical application of our main result (Theorem 2), it has numerous other (theoretical) applications such as NNGP correspondence, convergence to a kernel method in the limit of infinite width, and the Free Independence Principle, see Section 3 and Appendix B.
>
> "The proposed theory helps to explain some previous practices in learning neural networks; however, it does not suggest new techniques nor has empirical verification."
>
> While our theory does not suggest any new techniques, it demonstrates that the previous results are universal, meaning that exactly the same techniques (e.g. for hyperparameter tuning) should work the same way for all weight initializations - a result which is, as we believe, not obvious.
> We empirically validate some of the applications of our Theorem 2, including NNGP correspondence and convergence of the initial NTK; see Appendix I in the revised version of the manuscript. We plan to add more in future revisions.
>
> [1] Yang, G. (2019). Wide feedforward or recurrent neural networks of any architecture are gaussian processes. Advances in Neural Information Processing Systems, 32.
>
> [2] Yang, G., Hu, E. J., Babuschkin, I., Sidor, S., Liu, X., Farhi, D., ... & Gao, J. (2022). Tensor Programs V: Tuning Large Neural Networks via Zero-Shot Hyperparameter Transfer. arXiv preprint arXiv:2203.03466.

---

### Official Review · Reviewer_irWk · 2022-07-12

**Rating:** 6
**Confidence:** 2
**Soundness:** 4 excellent
**Presentation:** 2 fair
**Contribution:** 3 good

**Summary:**

The authors generalize the Master Theorem of Tensor Programs (TP) by demonstrating that non-Gaussian initializations result in many of same consequences of TP including (perhaps most interestingly) the NNGP correspondence (infinitely wide neural networks behave as Gaussian processes). Differences between this result and the original Gaussian initialized TP include the following: requirement of smooth nonlinearities, and weaker convergence (in probability versus almost surely). Much of the manuscript is dedicated towards proof.

**Questions:**

1. Why do the authors consider it useful to note that the semi-circle distribution and Marcenko-Pastur laws for non-Gaussian matrices fall out of Theorem 2? Going through tensor programs is not necessarily easier than existing proofs. What perspective/insights might deriving these results from TP have to offer?

**Limitations:**

Limitations and societal impacts are discussed.

**Strengths And Weaknesses:**

Strengths:
-The formalism introduced by TP was (and still is) of both theoretical and practical use (say, for hyperparameter tuning). Generalizing it even further is beneficial for the DL community.
-Proof of the non-Gaussian Master Theorem is thoroughly detailed.
-Nearly all of the consequences of the Gaussian Master Theorem are obtained.

Weakness:
-Notation is cumbersome. This is a challenging paper, indeed. Tensor Programs I (G. Yang) expends serious effort towards the development of NETSOR to aid the architectural universality result, but this paper is at times a pure exercise in analysis. Any type of diagram or visual to accompany the proof of Theorem 2 would have been nice.
-Requirement of smooth nonlinearities (although this is addressed, several times over).

---

> ### Author Response · Authors · 2022-08-01
> **Response**
>
> We thank the anonymous reviewer for valuable comments!
> We start with addressing the question:
>
> "Why do the authors consider it useful to note that the semi-circle distribution and Marcenko-Pastur laws for non-Gaussian matrices fall out of Theorem 2? Going through tensor programs is not necessarily easier than existing proofs. What perspective/insights might deriving these results from TP have to offer?"
>
> This is definitely not the central result of the paper, that’s why it is in the appendix. Still, while it is not necessarily easier to obtain the semi-circle and Marchenko-Pastur distributions using the TP formalism, the point was to demonstrate that this formalism is general enough to at least re-derive these classical results. More importantly, using the same method, the TP formalism allows to prove the Free Independence Principle for non-Gaussian TPs (in particular, for non-Gaussian-initialized neural nets) - the result which is, to the best of our knowledge, novel, and could hardly be obtained with the same proof methods as for the above-mentioned classical results. We discuss this result in Appendix B; it generalizes the original Gaussian result of TP III [1].
>
> Concerning the weaknesses noted by the reviewer:
>
> "Notation is cumbersome."
>
> In the latest revision, we have added more clarifying remarks, improved notation, and got rid of the most of the multiline formulas in the main.
>
> "Any type of diagram or visual to accompany the proof of Theorem 2 would have been nice."
>
> While we agree with this comment, it is not obvious for us how to visualize the proof of the main theorem at the moment.
>
> "Requirement of smooth nonlinearities."
>
> Smoothness requirement is strictly necessary for our proof method to work (it relies on Taylor expansion). Relaxing it seems challenging; it could constitute a good direction for future research.
>
> [1] Yang, G. (2020). Tensor programs iii: Neural matrix laws. arXiv preprint arXiv:2009.10685.

---

> > ### Comment · Reviewer_irWk · 2022-08-04
> > **Thanks for your response**
> >
> > I appreciate the authors cleaning up the notation and thank you for addressing my questions. I think the central result here is a nice addition to deep learning theory, but more of a complement to existing tensor programs theory rather an entirely novel perspective. This is a fairly niche result that will probably not affect a large audience, is valuable, nonetheless. Consequently, I keep my rating the same, "6: Weak Accept: Technically solid, moderate-to-high impact paper, with no major concerns with respect to evaluation, resources, reproducibility, ethical considerations."

---

### Official Review · Reviewer_3QrM · 2022-07-12

**Rating:** 6
**Confidence:** 2
**Soundness:** 4 excellent
**Presentation:** 3 good
**Contribution:** 3 good

**Summary:**

The paper generalizes the Tensor Programs framework, which computes infinite-width limits of deep networks / differentiable computation graphs under the assumption that weights are randomly initialized from independent Gaussians with variance 1/n, to the case of iid non-Gaussian initializations.

**Questions:**

nits:
- line 24: 'applications of Law of Large Numbers' doesn't parse, I would say 'applications of *the* Law ...'
- line 41: typo 'identially' -> identically


**Limitations:**

The technical conditions and limitations of the result are clearly articulated and discussed.

**Strengths And Weaknesses:**

The result in this work is an incremental extension of the Tensor Program framework to non-Gaussian weight initializations, and is clearly not intended as a standalone exposition of the framework. Unfortunately I have only a vague familiarity with the existing Tensor Program line of work and so can only provide superficial comments, not critique the result in any detail.

Significance: this result contributes to the theory of infinite-width limits of neural nets, which is certainly of general interest and has motivated interesting new methods that may be practical in some settings, e.g., for hyperparameter search. The specific contribution of this paper is incremental, and the result (although apparently nontrivial) does not seem particularly surprising, so its broader significance seems limited at best. Does this extension 'deserve' to be its own top-conference paper, rather than (say) an appendix to the general framework? I don't feel qualified to judge, but given the community's recent interest in this line of work I'm willing to give it some benefit of the doubt.

Quality and clarity: the paper is generally well-written. Technical conditions and results are clearly stated, as are the limitations and the relationship to previous results. I did not attempt to check the proof (section 4) but it appears well-motivated, structured and carefully reasoned; I believe it is likely correct.

Originality: the proof technique appears to follow a similar approach used in other recent work (Chen and Lam 2021), but its application and extension within the Tensor Programs framework is novel and, I think, nontrivial.

---

> ### Author Response · Authors · 2022-08-01
> **Response**
>
> First of all, we thank the reviewer for valuable comments!
> Below, we address the main concern posed by the reviewer.
>
> “The specific contribution of this paper is incremental, and the result (although apparently nontrivial) does not seem particularly surprising, so its broader significance seems limited at best.”
>
> While our results may not seem surprising when they are already present, we doubt these results (distribution universality) were apparent beforehand. Indeed, to the best of our knowledge, all empirical results supporting theoretical claims on infinitely wide nets applied Gaussian initialization as theory required it. However, our non-Gaussian Master theorem (Theorem 2) has the same corrolaries as the previous Gaussian one (Theorem 1), meaning that the previous theory can be readily and safely applied.
>
> Moreover, we would like to underline a small subtlety in our results that may seem surprising.
> Note that the limit in Theorem 2 does not change only if we replace the distributions of matrix entries with non-Gaussian ones, but this is not the case for the distribution of vector entries (Setup 2 requires them to be Gaussian). We can still model non-Gaussian vectors by applying a nonlinearity to Gaussian ones; that’s what we had in mind while considering the applications to neural nets, see Section 3 and Appendix A. However, swapping Gaussian vectors with non-Gaussian ones with a nonlinearity requires modifying the program, meaning that a program expressing computations in a Gaussian-initialized neural net and a program for the corresponding non-Gaussian initialized neural net can be different. Therefore they could have different inifnite-width limits.
>
> Still, Theorem 2 works and the limit exists in both cases, that’s why we do have NNGP correspondence and convergence to a kernel method (Corrolaries 1 and 2), but the distribution of vectors (e.g. biases) could affect the kernels.
>
> This means that the limit predicted by the Master theorem is not generally invariant wrt the distribution of vector entries. We demonstrate it in our experiments with a GRU network that converges to different NNGPs when bias vectors are Gaussian and non-Gaussian; see Appendix I in the revised version of the manuscript. We have also added a small remark in the main.

---

### Meta-Review · Area_Chair_oiRC · 2022-08-27

**Recommendation:** Accept
**Confidence:** Less certain

**Metareview:**

This submission is borderline.  Reviewers were generally in consensus --- all felt that the theoretical contribution is sound and non-trivial, but delivers a relatively minor addition to the Tensor Programs framework.  I fully agree with this perspective, and similarly to all reviewers, recommend that the paper be accepted and the NeurIPS community will be the judge of how significant its contribution is.

**Award:**

No

---

### Decision · Program_Chairs · 2022-09-14

Accept